# IMPROVING ADAM OPTIMIZER

**Ange Tato & Roger Nkambou**
Department of Computer Science
Université du Québec à Montréal
Montral, Qubec, Canada
{`nyamen_tato.ange_adrienne,nkambou.roger`}@uqam.ca

## ABSTRACT

We present a modified version of the Adam (Adaptive moment estimation) optimization algorithm, able to improve the speed of convergence and finds a better minimum for the loss function com-pared to the original algorithm. The proposed solution borrows some ideas from the momentum based optimizer and the exponential decay technique. The current step size made by Adam to update the parameters is modify in such a way that the new step takes in consideration the direction of the gradient and the previous steps update. We conducted several tests with deep Convolutional Neural Networks in the MNIST data. The results showed that AAdam(Accelerated Adam) outperforms Adam and NAdam (Nesterov ac-celerated Adam). The preliminary evidence suggests that making such a change improves the speed of convergence and the quality of the learned models.

## 1 INTRODUCTION

There are several ways to improve learning in deep learning neural networks such as improving the architecture (for example by making it deeper), finding the optimal parameters, playing with the data representation, choosing the best optimization algorithm etc. To date, there are no guidelines for setting up an optimal deep learning architecture. In this paper, we will be interested in a way to optimize the learning process in deep learning architecture using optimization algorithms based on the gradient descent method. The main goal is to get a better solution as quickly as possible.

The goal of an optimizer is to minimize an objective function (generally called the loss), which is intuitively the difference between the predicted data and the expected values. The minimization consists of finding the set of parameters of the architecture that give best results in the targeted tasks such as classification, prediction or clustering. There are several optimization algorithms in the literature Ruder (2016). The majority of these algorithms are first order methods and are based on the gradient descent method. Gradient descent techniques,e.g. back-propagation of error, are usually used to find a matrix of weights that meets error criteria. Adam (Adaptive Moment estimation) Kingma & Ba (2014) is probably the most used optimizer algorithm in the literature Goodfellow et al. (2016) Isola et al. (2017) Xu et al. (2015). In this paper, we propose a version of this algorithm called AAdam (Accelerated Adam) that improves the convergence of the algorithm in deep architectures (especially convolutional neural networks) and gives them a better ability to generalize on new data. AAdam compares favorably to other optimization methods. We conducted several tests with simple feedfoward neural nets and deep Convolutional Neural Networks in the MNIST [1] dataset. The results showed that AAdam outperforms not only Adam, but also its improved version known as NAdam (Nesterov Accelerated Adam) Dozat (2016). The preliminary evidence suggests that such a change improves the speed of convergence and the quality of the learned models.

## 2 RELATED WORK: ADAM AND NADAM

Learning in neural networks is done by minimizing an error function also called the loss function. This function therefore measures the difference between the expected outputs and calculated on the complete sample. An error close to 0 implies that the network correctly classifies the data on

---

[1]http://yann.lecun.com/exdb/mnist/

which it has learned. However, recovering the global minimum becomes harder as the network size increases and this is in practice, irrelevant as global minimum often leads to overfitting Choromanska et al. (2015). Thus, the goal is to minimize the cost but by making sure to not overfit the model on training data. The ideal being to lead to a network able to well classifying the data used for training as well as the data they have never seen before (validation data). AdamKingma & Ba (2014) is a first order gradient based algorithm of stochastic objective functions, based on adaptive estimates of lower-order moments. The first moment normalized by the second moment gives the direction of the update. Adam updates are directly estimated using a running average of first and second moment of the gradient. It computes adaptive learning rates for each parameters. In addition to storing an exponentially decaying average of past squared gradients $v_t$ like AdaDelta and RMSprop, Adam also keeps an exponentially decaying average of past gradients $m_t$, similar to momentum. NAdam Dozat (2016) is a modifyed version of ADAM momentum that takes advantage of insights from NAG(Nesterov accelerated gradient).

## 3 AADAM: ACCELERATED ADAM OPTIMIZER

The main idea behind AAdam is to speed up the progress along dimensions in which gradient consistently point in the same direction. In addition to storing an exponentially decaying average of past squared gradients $v_t$ and an exponentially decaying average of past gradients $m_t$ like Adam, AAdam also keeps an exponentially decaying average of past updates. Thus, the current update not only depends on the previous gradients, it also depends on the previous values of the update $\Delta\theta$ . We keep track of past parameters updates with an exponential decay where $\beta_1$ (approximatively 0.9, the same $\beta_1$ in Adam) is the constant controlling the decay. It adds a small value $d$ to the current update of Adam. That value ($d$) is multiplied by the sign of the current gradient. The new update rule is summarized as follows :

$$
\begin{aligned}
\theta_{n+1} &= \theta_n - (\eta \frac{\beta_1}{\sqrt{\hat{v_n}} + \epsilon} \hat{m_n} + d) \\
d &= \Delta\theta_{n-1} * sign(\nabla_{\theta_n} J(\theta)) * (1 - \beta_1) \\
\hat{m_n}, \hat{v_n} &= \text{Adam Parameters} \\
\Delta\theta_{n-1} &= \text{Last update step.}
\end{aligned}
$$

**Intuition behind AAdam** : If we consider that our objective is to bring a ball (parameters of our model) to a lowest elevation of a road (cost function), what we do is to adapt the speed of the ball by trying to sending it more in the direction of the gradient. That also implies decreasing the step size taken by the ball on the opposite direction. This is done by adding a small portion of past updates to the current updates of Adam. The update is a vector that has the direction of the gradient. In case the gradient changes direction, the size of the step taken by AAdam will be less larger than the one taken by Adam step. This new update accelerate the move of the ball towards the minimum (local or global depending on where we started). Since the step added to the step proposed by Adam is not very big, one can hope that if Adam finds a better minimum, AAdam will find it too but more quickly. It should not be forgotten that finding a better minimum does not imply a better ability to generalize, on the contrary, finding a better minimum could lead to overfitting.

## 4 EXPERIMENTS

In order to empirically evaluate the proposed modification, we investigated deep convolutional neural networks. Using the MNIST dataset, we demonstrate promising results. We use the same parameter initialization when comparing AAdam with Adam and Nadam. The CNN has two layers of convolution and two fully connected layers. The best learning rate found for those optimizers was 0.002, $\beta_1$ was set to 0.9 and $\beta_2$ was set to 0.999 as recommended for Adam Kingma & Ba (2014). No pre-processing was applied to training images. All optimizers were trained with a mini-batch size of 250. All weights were initialized from a values truncated normal distribution with standard deviation 0.1. The biases values were initialized to 0.1. We used the tensorflow library which already proposes the implementation of Adam optimizer. Since we did not find any available implementation of NAdam optimizer, we developed it by our selves using python and on the basis of the implementation

of Adam available in tensorflow. We made the code of AAdam and NAdam available and on our github repository [2].

## 5 RESULTS AND DISCUSSIONS

We conducted several tests where we changed the CNN architecture (deep to non-deep) and the different parameters; AAdam gave better results in the majority of cases. We chose to show only a part of these results in the graphs. Figure 1-1 shows the evolution of the loss value on the training data. The results in this figure show that the three optimizers have practically the same performance. They yield similar convergence. We slightly cropped the figure, but at the five thousandth step, AAdam gives the lowest value of the loss which was $2.208 * 10^{-4}$ compared to respectively $3.019 * 10^{-4}$ and $2.921 * 10^{-4}$ for Adam and NAdam. NAdam is nevertheless the most faster. As we can see, AAdam is between Adam and NAdam most of the time. In figure 1-2, AAdam outperforms the other two optimizers on validation data by a relatively large margin. The empirical performance of AAdam is consistent with the intuition behind the method. We can see that AAdam has a better generalization capacity than the other two algorithms. Between the step 3000 and 4000, when the loss value of NAdam and Adam is increasing, the loss value of AAdam is decreasing. This is a good behavior since the objective is to minimize the cost on the validation data as much as we can. We can also notice that, even if NAdam has a better convergence compared to other two on the training set, it has the worst behaves on the validation set. The best accuracy so far we had on the 10000 images test data of MNIST for ADAM was 99.12, for NAdam 99.08 and for AAdam 99.28.

In summary, AAdam achieves the best results especially on the validation set. Even if it requires more memory, AAdam clearly outperforms Adam and NAdam in reducing training and validation loss. AAdam also gives better accuracy than the others.

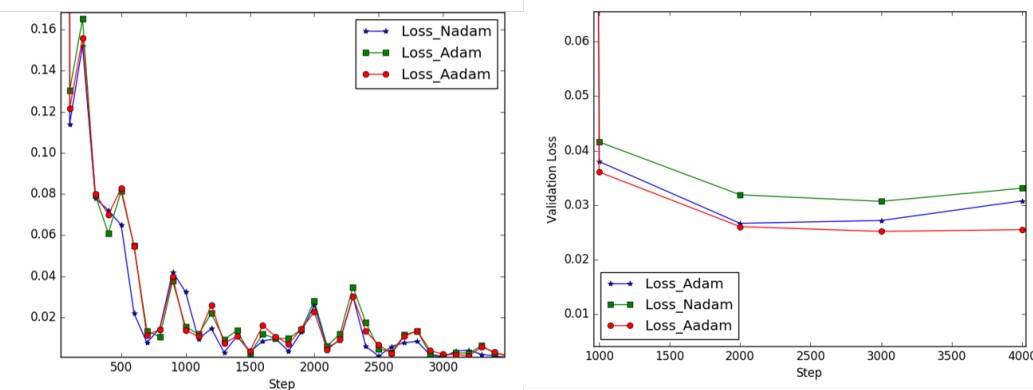

Figure 1: 1) The variation of the loss value in the training data. AAdam is between Adam and NAdam most of the time. 2) The variation of the loss value in the test data. AAdam outperforms Adam and NAdam with same settings. The validation data consist of 10000 images.

## 6 CONCLUSION

In this paper, we introduced a simple and intuitive method to modify Adam optimizer and to make it more efficient. This work takes Adam algorithm one step further, and improves its convergence and its generalization without noticeably increasing complexity. The only drawback of the proposed solution is that it takes more memory than the standard approach. In one hand, if one care much about 'very' fast convergence and less on the generalization ability of the final model, one should choose NAdam. On the other hand, if one care about fast convergence (not as much as with NAdam) but wants a more robust model able to generalize well on new cases, one should choose AAdam.

---

[2]https://github.com/angetato/Optimizers-for-Tensorflow

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
