# OpenReview forum: "IMPROVING ADAM OPTIMIZER"
_ICLR.cc/2018/Workshop — Reject_

### Official Review · AnonReviewer3 · 2018-03-06
**Insufficient experiments and no theory**

**Rating:** 4
**Confidence:** 4

**Review:**

Summary:
This paper suggest ,AAdam, an alternation of the Adam optimizer that incorporates an additional term which encourages progress in directions where
the gradients consistently point in the same direction. Then they show empirical results comparing their performance to Adam and another accelerated version of Adam.

The Algorithm:
The authors suggest a simple modification. However, this modification uses the sign of the gradient estimates which may result a very bad behaviour even for simple convex problems. The authors do not discuss this issue in their paper.
Also, the authors do not provide any theoretical guarantees.

Experiments:
The authors only make comparisons for the MNIST dataset, which is insufficient.
Illustrating the benefits of AAdam requires a much more elaborate experimentation.

I therefore recommend to reject the paper.

---

### Official Review · AnonReviewer1 · 2018-03-08
**Interesting idea; too little content**

**Rating:** 3
**Confidence:** 4

**Review:**

The paper presents an interesting modification of Adam. The submission is low on content however, even for a workshop paper. Some basic intuition for the idea is provided, but no more than that. Performance is evaluated on a *SINGLE* experiment on MNIST, where the AAdam slightly outperforms Adam and Nadam.

---

### Official Review · AnonReviewer2 · 2018-03-09
**Hyperparameters study is missing**

**Rating:** 3
**Confidence:** 5

**Review:**

The empirical results shown in the paper are not convincing. The tiny difference shown on MNIST could be due to a particular choice of hyperparameters or even random seed (see the bumps). For instance, the difference of AAdam and Adam might be roughly  approximated by a different choice of momentum coefficients and/or baseline learning rate in the original Adam.

---

### Decision · Program_Chairs · 2018-03-20
**ICLR 2018 Workshop Acceptance Decision**

**Decision:**

Reject

**Comment:**

Based on the reviews, this paper has not been accepted for presentation at the ICLR workshop. However, the conversation and updates can continue to appear here on OpenReview.